# Development of an Inactivated Camelpox Vaccine from Attenuated Camelpox Virus Strain: Safety and Protection in Camels

**DOI:** 10.3390/ani13091513

**Published:** 2023-04-30

**Authors:** Kuandyk Zhugunissov, Muratbay Mambetaliyev, Nuraiym Sarsenkulova, Shalkar Tabys, Marzhan Kenzhebaeva, Arman Issimov, Yergali Abduraimov

**Affiliations:** 1Research Institute for Biological Safety Problems, Gvardeiskiy 080409, Kazakhstan; 2Department of Biology, K.Zhubanov Aktobe Regional University, Aktobe 030000, Kazakhstan; 3Department of Otolaryngology, University of Iowa, Iowa City, IA 52242, USA

**Keywords:** camelpox, virus, inactivated vaccine, safety, protection

## Abstract

**Simple Summary:**

Camelpox is an important infectious viral disease of camels. This viral infectious disease is considered one of the major concerns for camel breeding countries, and the most effective way to solve this problem is to develop a vaccine. In this research, we developed a vaccine against camelpox. Our developed vaccine was tested in mice and camels, and it was found that the candidate vaccine was innocuous to mice and camels. This developed vaccine can be used in camel breeding farms in the future to prevent significant economic losses caused by camel pox.

**Abstract:**

This article describes the preparation of an inactivated vaccine from an attenuated strain of camelpox. The attenuated camelpox virus (CMLV) was grown in lamb kidney cells and in Vero cells. CMLV was accumulated to a significantly higher (*p* ≤ 0.05) titer in lamb kidney cells (7.75 ± 0.08 log TCID_50_/_mL_) than in Vero cells (4.00 ± 0.14 log TCID_50_/_mL_). During virus inactivation, a concentration of 0.05% beta-propiolactone (BPL) completely inactivated the virus in 6 h at a temperature of 22 ± 1 °C, while a concentration of 0.2% formaldehyde inactivated the virus in 8 h. However, a viral antigen inactivated by BPL was used for vaccine preparation. The inactivated viral antigen was adsorbed with aluminum hydroxide gel, and as a result, an inactivated candidate vaccine was prepared. While the safety of the candidate vaccine was tested in camels and white mice, the protective efficacy of the vaccine was tested only in camels. In the safety evaluation of the inactivated vaccine, the vaccine was not observed to cause any adverse effects in mice and camels. During the immunogenicity study in camels, antibody formation started (0.2 ± 0.16 log2) at Day 21 post-vaccination (PV), and the antibody titer peaked (1.33 ± 0.21 log2) at Day 60 PV and decreased at Day 90 PV (0.50 ± 0.22 log2). Furthermore, no antibodies were detected in vaccinated camels from Days 180 to 365 PV. Camels that received vaccination and were subsequently exposed to wild-type virus evinced a healthy state despite lacking antibodies. In contrast, unvaccinated camels exhibited susceptibility to camelpox upon challenge.

## 1. Introduction

Camelpox is a contagious viral disease characterized by fever, head swelling, and the appearance of papular eruptions on the skin and mucous membranes of diseased animals. It is also characterized by abortion in female camels and death of the colts. The camelpox virus (CMLV) belongs to the *Orthopoxvirus* genus, a member of the vast *Poxviridae* family, which includes humans, other mammals, fish, reptiles, and invertebrates. Camels of all ages are susceptible to this poxvirus, but young stock are stricken more often and have more severe symptoms [1]. Camelpox outbreaks have substantial economic consequences in herds, as affected camels may suffer from weakness and reduction in milk production and weight [2]. In addition, the occurrence of camelpox in herds may favor secondary infections from other circulating diseases from which camels might die [3].

Camelpox has been recorded in the Middle East (Iran, Iraq, Saudi Arabia, United Arab Emirates (UAE), and Yemen), in Asia (India, Afghanistan, and Pakistan), in Africa (Algeria, Egypt, Kenya, Mauretania, Niger, Somali Morocco, Ethiopia, Oman, and Sudan), and in Central Asia (Kazakhstan and Turkmenistan) [4]. The last outbreak of camelpox in Kazakhstan was in 1996 in three districts of the Mangystau oblast. On 8 farms of the Mangystau oblast with 8000 camels, 830 were infected, and 43 of those died of the infection [5]. Following this outbreak, new laboratory-confirmed cases of camelpox were reported in the Mangistau Region in the summer of 2019 [6].

The main tool for controlling camelpox is targeted prophylaxis. Currently, four camelpox vaccines are available worldwide [1], two of which have been evaluated and commercialized. However, the lack of commercial vaccines in many camel-rearing countries is a major hindrance to controlling camelpox. These commercial vaccines have never been used in Kazakhstan.

The advantage of live vaccines due to manufacturability and ease of use is beyond doubt. However, at the same time, when such vaccines are administered, there is a risk of disease in highly susceptible individuals (the so-called post-vaccination complications). There is also a certain risk of isolating other vaccine strains. [7]. Cell lines or embryos can carry a latent viral infection that can have a detrimental effect on the vaccinated organism when contaminated with a live vaccine. For example, bluetongue virus was detected in live commercial vaccines against sheep-pox and lumpy skin disease [8].

Moreover, farmers in the southern and southeastern regions of Kazakhstan oppose the use of live vaccines, although live vaccines previously developed by the Research Institute for Biological Safety Problems (RIBSP) [9] have been used successfully in the western regions of Kazakhstan, where camelpox is common (unpublished data). Since camelpox had never previously been observed in the southern and southeastern regions, farmers believe that the use of a live vaccine would cause camelpox outbreaks. Because of this belief, the use of a live vaccine in prosperous regions was not supported by farmers. In this regard, taking into account the well-being of the southern regions of Kazakhstan (since these regions of Kazakhstan are camelpox-free zones) where camel breeding is intensively practiced in parallel with the live vaccine, a safe inactivated vaccine based on an attenuated strain of the virus was developed to control outbreaks and reduce viremia and virus circulation throughout this environment. Thus, the current study demonstrates the results from the development of the vaccine itself to the study of its protective effects in target animals.

## 2. Materials and Methods

### 2.1. Virus

In the present study, the KM-40 strain of the CMLV was used. The virus was obtained from the parental strain M-96 after 40 passages in chicken embryos of 11 to 12 days of age. The KM-40 strain was obtained from the Republican Depository of Especially Dangerous Pathogens of the Research Institute for Biological Safety Problems. The infectious activity of strain KM-40 was 6.25 ± 0.08 lg EID_50_/_mL_.

The virulent wild-type strain M-96 (GenBank # AF438165.1), isolated from diseased camels during a field outbreak recorded in the Mangistau region of Kazakhstan in 1996, was also used for the challenge study [10,11]. The infectious activity of virulent strain M-96 was 5.75 ± 0.14 lg EID_50_/_mL_.

### 2.2. Animals

The study utilized camels from two species, namely *Camelus bactrianus* and *Camelus dromedaries*, which were sourced from farms situated in the southern region of Kazakhstan. The camels were between 10 to 18 months old, and they were screened for acute infectious diseases and found to be negative. Additionally, they tested seronegative for CMLV antibodies. Prior to experiments, animals were kept in quarantine for four weeks. During the quarantine period, blood samples were tested for the presence of antibodies against CMLV using the serum neutralization test (SNT).

In the experiment, animals that did not have specific antibodies to the CMLV were used. Animals were kept in special rooms, and access to food and water was provided ad libitum.

### 2.3. Animal Ethics

This study was performed in compliance with national and international laws and guidelines on animal handling, and the experimental protocol was approved by the Committee on the Ethics of Animal Experiments of the RIBSP of the Science Committee of the Ministry of Education and Science of the Republic of Kazakhstan (permit number: 0818/021 and 0105/022).

### 2.4. Vaccine Candidate Preparation

Lamb kidney cells and Vero cells were cultured in Dulbecco’s modified Eagle’s medium (DMEM) with 10% fetal calf serum at 37 °C. Cell monolayers were formed within 1 day after cell seeding. Subsequently, cell medium was aspirated, and confluent cell monolayer was infected with the virus at different doses (multiplicity of infection (MOI): 0.1, 0.01, 0.001, and 0.0001). Then, infected cells were incubated at 37 °C for 1 h. After adsorption for 1 h, DMEM with 2% fetal calf serum was added to the infected cells. Virus-infected cells were further incubated at 37 °C for 7 days [12]. When 80–90% of the cell monolayers exhibited 80–90% CPE, they were frozen at −40 °C for 16–18 h. Then, virus suspension was thawed at 20 ± 3 °C and subsequently inactivated with β-propiolactone (BPL) and formaldehyde using various concentrations (Sigma-Aldrich, St. Louis, MO, USA) at temperatures of 22 ± 1 °C and 37 ± 0.5 °C [13]. Prior to inactivation, the viral suspension was frozen and thawed three times at –40 °C and centrifuged to eliminate cell debris at 1000 rpm for 10 min. Following centrifugation, the supernatant was inactivated using inactivating agents mentioned above. After inactivation, BPL was inactivated using sodium thiosulfate solution, and formaldehyde was neutralized by adding sodium bisulfite solution at a final concentration of 0.25%. The degree of virus inactivation was assessed by inoculating cell cultures and serially passaging the samples three times. Finally, the inactivated and purified viral antigen was combined with an Al(OH)3 adjuvant and incubated overnight at a temperature range of 4 to 6 °C to allow the absorption of the viral antigen.

### 2.5. Safety

The safety of the vaccine was tested on 20 white mice and 6 camels. The experimental animals were divided into two groups of equal numbers. White mice (n = 10) were injected with the vaccine intraperitoneally in a volume of 0.2 mL, and camels (n = 3) were injected intramuscularly in a volume of 10 mL. Both animal species in the placebo group (white mice (n = 10), camels (n = 3)) were injected with phosphate buffered solution at the same volume and using the same method, depending on the animal species, and were considered the control group. Animals were monitored daily for 14 days, and the presence of clinical signs was recorded.

### 2.6. Immunogenicity

Six camels were injected intramuscularly with the first dose (5.0 mL) of the inactivated camelpox vaccine candidate in the left side of the neck. On Day 35 after administration of the first dose, the same volume of second dose was administered on the same side. After introduction of the first and second doses of the vaccine, the animals were monitored for their general condition for 14 days with daily body temperature measurements. In addition, on Days 7, 14, 21, 28, 35, and 42 and every month after the first dose administration, blood samples were collected from the vaccinated camels to measure the dynamics of the formation of virus-neutralizing and specific antibodies using a neutralization test (Appendix A).

### 2.7. Challenge Study

Animals were challenged at 180 and 365 days after vaccination (Figure 1). For this challenge, 12 camels were utilised, 6 of which were vaccinated, and the remaining 6 were used as controls. At each point of the challenge study, three vaccinated camels and three control camel were challenged. The virulent strain “M-96” of CMLV (GenBank # AF438165.1), isolated from diseased camels during an outbreak of camelpox in 1996 in Kazakhstan, was used for the challenge study. The strain is recommended as a virulent control for evaluating the immunogenicity of camelpox vaccines [10]. The lyophilized virulent strain M-96 was obtained from the laboratory of the Collection of Microorganisms and was reconstituted to its original volume with 50% glycerol and injected by scarification into the hairless area of the hind limbs of camels at a dose of 10^5.0^ EID_50_/0.2 mL. Animals were monitored for 14 days, with attention to the general condition, temperature, and clinical signs characteristic of camelpox.

The immunogenicity of the vaccine was evaluated in animals by means of a challenge using the wild-type strain “M-96” of the camelpox. At the same time, the virulent strain “M-96” of the CMLV was administered to vaccinated animals by scarification at a dose of 10^5.0^ EID_50/_0.2 mL. Recipient animals were observed daily for clinical manifestation of camelpox. During clinical observation, attention was drawn to the general condition of the animals (suppressed appetite, lethargy, the presence or absence of papules, and generalization of the process). At the same time, vaccinated animals should not have demonstrated any clinical signs characteristic of camelpox, while unvaccinated animals should have shown an increase in body temperature, hyperemia, papules at the inoculation sites, and development of generalized poxvirus infection.

### 2.8. Serum Neutralization Test (SNT)

Neutralizing antibodies to CMLV were detected in a serum–virus neutralization test (SNT) with the constant virus, varying serum method. SNT were carried out in 96-well cell culture plates as described previously by OIE protocol (2012) [14] using a virulent CMLV strain and normal camel serum. The sera were diluted from 1:2 to 1:128 and mixed in equal volumes with the CMLV, which was used at a dose of 100 TCID50/mL. The serum–virus mixtures (equal volumes of 100 μL) were incubated at 37 °C for 1 h before the introduction to LK cells (3 to 5 × 104 cells/well/100 μL). The plates were incubated in a humidified atmosphere containing 5% CO_2_ at 37 °C for 120 h. The final reading was conducted on the basis of the presence or absence of cytopathic effect (CPE). For virus neutralization, wells were scored as positive if 100% of the cell monolayer remained intact. The maximum serum dilution that leads to complete virus neutralization (without cytopathic effect) in 50% of the wells during the test is considered as the 50% end-point titer of the serum. In other words, the final serum dilution that inhibits 50% of the cytopathic effect of the CMLV is determined as the VNT titer (Appendix A).

The monolayers were examined daily for specific CPE by inverted microscopy and end-points calculated according to Reed and Muench [15].

### 2.9. Delayed Type Hypersensitivity Test (DTH)

A delayed type hypersensitivity test (DTH) was performed according to the procedure [16] using the antigen prepared from the CMLV strain grown in an LK cell culture. Briefly, when the cytopathological effects of the virus on the monolayer reached 80–90% after inoculation with the CMLV strain into the LK cell culture, the contents of the vials were collected, subject to three cycles of freeze–thaw, and centrifuged at 1500 rpm for 10 min. Next, the supernatants were collected and aliquoted in 1 mL samples and stored at 4 °C. Prior to use, the antigen was inactivated by heating at 56 °C for 1 h, and this antigen was used as the antigen for DTH.

All vaccinated and unvaccinated camels were intradermally injected with 0.2 mL of heat-inactivated CMLV antigen into a shaved area on the left side of the neck. The thickness of the skin at the site of inoculation was measured every other day for five days as an indicator of the hypersensitivity reaction as measured using a caliper.

### 2.10. Statistical Analyses

Statistical analysis of the research results was performed using the GraphPadPrism 9 program (GraphPad Software, Inc., La Jolla, CA, USA). Descriptive statistics were applied to all data. Mean values (M) and standard deviations (SD) were calculated. Differences between infectious titers obtained with LK and Vero cells as well as between antibody titers in vaccinated and unvaccinated animals were determined using a one-way analysis of variance (ANOVA) followed by a Student’s *t*-test. Values of *p* ≤ 0.05 were considered significant.

## 3. Results

### 3.1. Comparative Cultivation of the Virus in Cell Cultures Depending on the Multiplicity of the Infectious Dose (MOI)

The CPE induced by various doses of camelpox virus was directly related to the type of cell culture and infective dose (Figure 2a). Specifically, the destructive action of the virus in a monolayer of Vero cells was faster than that in LK cells, and the disaggregation of these cells was clearly observed (Figure 2b). However, the virus titer was higher in LK cells than in Vero cells at all tested doses (*p* ≤ 0.0001), except for MOI 0.0001. At this dose, the CPE was observed 72 h following inoculation, and the virus titer reached the maximum value after 120 h. The virus titer was 4.58 ± 0.57 log TCID50/mL in LK cells and 4.50 ± 0.25 log TCID50/mL in Vero cells. At the highest infective dose (MOI 0.1), virus accumulation was observed in a lower titer than at other doses. Further incubation resulted in a decrease in virus titer. Current experiment established that the camelpox virus accumulates in the highest titers at infective doses equal to MOI 0.01–0.001, with the virus titer in LK cell culture reaching 7.08–7.41 log TCID50/mL, whereas in Vero cell culture, it reached 3.91–4.33 log TCID50/mL.

### 3.2. Virus Inactivation with β-Propiolactone (BPL) and Formaldehyde

The virus was inactivated by the BPL solution at a final concentration of 0.05% in 5 h at 37 °C and in 6 h at 22 °C (as shown in Figure 3a). After exposure to 0.05% BPL, the virus inactivation was evaluated by checking for virus-associated CPEs in the cell culture monolayer over three passages, but none was detected. Virus inactivation using 0.2% formaldehyde aqueous solution at temperatures of 22 and 37 °C found that under both temperature conditions, the virus was inactivated by 8 h (Figure 3b). The viral suspension treated with 0.2% formaldehyde did not cause any CPEs in the cell monolayer, which indicates complete inactivation of the virus.

After obtaining inactivated CMLV antigens, we studied the antigenic activity of inactivated virus-containing antigens in goats. In this process, goats were immunized intramuscularly with inactivated materials with a volume of 3 mL. To determine the dynamics of the formation of VNA in immunized goats, blood samples were collected on Days 7, 14, 21, and 28 after immunization and tested using the SNT.

All studied materials treated with various concentrations of BPL and formaldehyde formed virus-neutralizing antibodies (VNAs) in immunized goats in titers ranging from 1.0 to 2.0 log2 on Days 21 and 28 after immunization (Figure 3c,d). Only the material inactivated with BPL at a final concentration of 0.05% at t° −22 ± 1 °C evoked an immune response seven days earlier than the other materials treated with BPL and formaldehyde. However, statistical analysis using the ANOVA method confirmed the absence of a significant difference between VNA titers detected on Days 21 and 28 post-vaccination (*p* ≥ 0.05).

Thus, based on the results, BPL was chosen as an effective inactivating agent, since BPL at a final concentration of 0.05% and temperature of 22 °C inactivated CMLV for 6 h, with maximum preservation of antigenic activity.

### 3.3. Vaccine Safety in Mice and Camels

The inactivated vaccine candidate did not cause any systemic adverse effects in the mice after administration. No signs of depression, loss of appetite, or body weight were observed (Figure 4a).

After administration of the vaccine to camels, no signs of the disease were detected, and the general condition of the animals was satisfactory. Upon palpation of the injection site, swelling was noted, which persisted in camels for 24 to 72 h, after which it disappeared on its own without causing any adverse effect. In addition, in vaccinated animals on Days 4 and 5 after vaccine administration, a slight increase in body temperature to 39.9 °C was noted, which returned to normal on Day 6 post-vaccine immunization (Figure 4b).

Neither type of animal in the control group manifested any signs of deviation from physiological norms, temperature increase, or local or systemic changes in the physiological parameters of the body after the introduction of saline.

### 3.4. Serum Neutralization Responses

The inactivated vaccine did not cause an immune response in camels on post-vaccination Days 7 and 14. The formation of VNA in the organisms of immunized camels was observed from Day 21 in only one camel of the nine immunized animals. On the 28th day, antibodies were found only in two of the nine vaccinated camels, and on the 35th day, antibodies were detected in four camels. On Day 35, the camels received a second dose of the vaccine.

Antibodies were detected in all camels 42 days after receiving the second dose of the vaccine in titers ranging from 1.0 to 2.0 log2. The peak of the VNA titer up to 2.0 log2 was observed on Day 60. However, on Day 90, VNAs were found in only three of the nine vaccinated camels. At other points (180, 270, and 365 days), no antibodies were detected (Figure 5).

### 3.5. Delayed Type Hypersensitivity Test (DTH)

The results of the DTH are shown in Table 1. All vaccinated camels responded positively to the hypersensitivity test regardless of vaccination time, while unvaccinated control camels inoculated with uninfected cell culture homogenate showed no increase in skin thickness at the injection site when testing. Twenty-four hours following inoculation with the warmed-up inactivated CMLV strain, a 2–3-fold increase in skin thickness was found in vaccinated animals, which then regressed on the fifth day post-inoculation.

## 4. Challenge Study

M-96 strain was utilized in the challenge experiments to study the resistance of vaccinated organisms to the wild-type of the CMLV. As a result of the studies, the vaccinated animals remained in a satisfactory condition, and their body temperatures remained within the physiological norm (38.0–39.8 °C) during the entire study period (30 days). At that time, the unvaccinated control animals manifested clinical signs characteristic of camelpox. At the same time, the body temperature of the animal increased to >40.0 °C for seven days, and papules developed at the virus injection site on Days 9–10 after infection. On Days 12–14, animals exhibited generalization of the disease, with development of firm nodules from 2 to 3 mm in size on the upper lip, in the neck, and on the skin of the inner side of the hind limbs in addition to an increase in pre-scapular and submandibular lymph nodes (Figure 6a–f).

## 5. Discussion

Vaccination is the most efficient tool to halt and control the spread of infection in endemic and recently affected areas. However, in the event of an outbreak, choosing the best vaccine is a major challenge for veterinary authorities and farmers [17,18,19]. In this regard, we believe that the development of an inactivated vaccine, in addition to a live camelpox vaccine, will contribute to solving this problem. A considerable amount of literature has been published on the safety and efficacy of live camelpox vaccines [9,16,20,21,22,23], while studies on inactivated camelpox vaccine [16] are rare. In our work, we discuss the results of the work conducted on obtaining the viral mass, inactivation, safety, and efficacy of the inactivated vaccine. In the literature, the development of inactivated vaccines against pox virus infections has often been unsuccessful [24]. Proof of the lack of success is associated with the inactivated camelpox vaccine [25] and the inactivated sheep-pox vaccine developed in the Soviet Union [24]. This situation requires special attention to the parameters of growth and virus inactivation to obtain a highly active viral antigen for the preparation of an inactivated vaccine. Therefore, the importance of obtaining highly active viral material cannot be efficiently addressed without improving the existing ones and determining the optimal growth parameters that ensure the production of viral suspension possessing a high titer. It should be noted that earlier, we used developing chicken embryos of 11 to 13 days of age to generate CMLV [9,26]. However, this growth system is laborious and material-intensive for preparation of an inactivated vaccine. The results of previous studies [12,27] made it possible to determine the optimal parameters for cultivating the attenuated CMLV strain in LK and Vero cell cultures. Several growth parameters (optimal growth temperature, serum concentration in the media, and volume of media) were obtained on the basis of previous studies [12,27], and in this experiment, we determined the dose of infection and the timing of virus cultivation in LK and Vero cell cultures (Figure 1). As a result, the optimal MOI for the KM-40 strain was 0.01–0.001 TCID_50_/cell, and the cultivation period for obtaining viral biomass with high infectious activity was 96 h. These results are consistent with the data obtained for the parent strain KM-40 [27]. However, Kutumbetov et al. reported that the cultivation of pox viruses in a sensitive primary trypsinized cell culture (LK, LT) promoted the preservation of the pathogen’s immunogenicity in prolonged passages, while in continuous cell lines, such as Vero, the virus underwent superattenuation after several passages and sharply reduced the immunogenic index [24].

Only the inactivated vaccine against camelpox is known [16]; however, the technology for manufacturing this vaccine is not available. In this regard, for the inactivation of CMLV, we focused on the results previously obtained by other researchers in the inactivation of pathogens of other poxvirus infections [28,29,30]. To develop the optimal parameters for the inactivation of the CMLV, we used BPL and formaldehyde, which are widely used in the development of inactivated vaccines. The advantages and disadvantages of these inactivating agents are detailed in a number of literature sources [31]. Both inactivants are known to be alkylating agents. However, their mechanisms of action on the structure of the virus are different. For example, formaldehyde affects both the genome and proteins, and BPL acts mainly as an alkylating agent on the guanine of viral DNA or RNA [31]. In this regard, we tried to determine the “gentle” mode of inactivation of the CMLV using these two inactivating agents. The purpose of the comparative experiment was to identify an inactivating agent that completely destroys the infectious activity of the virus but retains its epitope as much as possible, causing protective immunity from infection with the wild poxvirus. As a result of the study, both inactivating agents completely inactivated the virus and elicited an immune response in animals. However, among the tested inactivating agents and inactivation modes, we chose BPL at a final concentration of 0.05% at a temperature of 22 ± 1 °C for 6 h. The selected mode of inactivation was gentler and caused an immune response earlier than other modes of BPL and formaldehyde.

The main advantage of inactivated vaccines is their safety. The safety of the vaccine was tested by inoculating camels with a double dose (10 mL). This test was carried out in accordance with the OIE safety testing protocol for inactivated vaccines [14]. In addition, although not specified in the OIE protocol, we tested the safety of the vaccine in mice. As a result, the vaccine did not show any adverse reactions in these animals during a 14-day observation period. When studying the immune response in vaccinated animals after vaccination, it was found that partial formation of specific antibodies to the CMLV begins on the 28th day after the first immunization, but a 100% immune response occurred on the 25th day after the second vaccination. When studying the duration of immunity induced by the inactivated vaccine, specific antibodies to the CMLV were detected in the blood of vaccinated animals within 90 days after the second immunization, a finding that confirms the presence of humoral immunity in the body of immunized animals. However, starting at 90 days to 180 days after vaccination, the vaccinated animals showed a decrease in the dynamics of the formation of specific antibodies to a titer of 1:2, and even in some camels, no antibodies were found. Different results were obtained by other researchers during field trials of the inactivated vaccine in camels [16]. The difference between the field trial conducted by Khalafalla and El Dirdiri and our results is the detection of VNAs in vaccinated camels over one year, with titers ranging from 1:4 to 1:32. In our study, no CMLV antibodies were detected in vaccinated camels at 180 and 365 days. However, these camels remained healthy and were alive without becoming sick during the challenge with the wild-type virus. A possible explanation for this finding could be a decrease in the immune properties upon inactivation or attenuation of poxviruses due to a long passage in the biological system and a concurrent increase in paraspecific effects [32]. It is important to note that in poxvirus infections, cellular factors play a more prominent role as protective immunity factors. Humoral factors may be absent or present at low levels, which cannot be detected by available tests (SNT or ELISA). The protection of animals or the presence of immunity in such cases is confirmed by resistance to infection with virulent virus. [24].

Other studies have shown conflicting results regarding the protective properties of inactivated camelpox vaccines. For example, the inactivated vaccine developed in the former Soviet Union did not protect camels from a virulent virus [33], while the Moroccan inactivated vaccine had protective properties [16], suggesting that lack of protection of the Soviet Union vaccine is due to improper choice of substrate for obtaining a highly active viral antigen and effective parameters of virus inactivation, which is considered one of the most critical stages of vaccine technology [7,33].

In this case, the need arose to study cellular immunity; however, this parameter was not evaluated in the current study because the lack of commercial kits and reagents required for testing cellular immunity in camels hindered the study of post-vaccination cellular immunity in camels. However, we generally determined the presence of cellular immunity using an additional DTH test. Studies using this method are widely used in other poxvirus infections [34,35]. In our study, it was noted that two months after camels were vaccinated and tested with a slow type of hypersensitivity, the skin thickness increased by 2 to 3 times, whereas after 6 and 12 months of testing post-vaccination cellular immunity in camels with a slow type of hypersensitivity, the skin thickness was increased only 1.5–2 times compared with control animals.

After analyzing the results, it was found that the inactivated vaccine developed from the attenuated strain was safe for camels aged 10 months and older. However, although the inactivated vaccine elicited a very weak immune response in camels, it was highly effective during the challenge study with the virulent strain. In this regard, although it is known that cellular immunity is more important than humoral type of immunity during poxvirus infection [36,37], it is of importance to examine the post-vaccination cellular immunity against camelpox infection in camels in the future.

## Figures and Tables

**Figure 1 animals-13-01513-f001:**
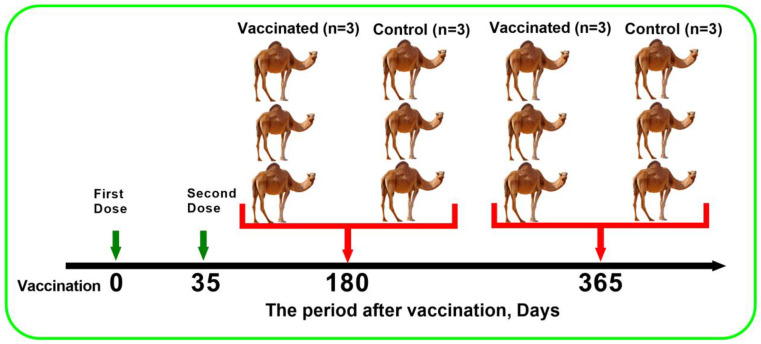
Design of the study on the protectiveness of an inactivated camelpox vaccine. The green arrow shows the number of days post-vaccination of the first and second doses. Camels were challenged 180 and 365 days after inoculation with the second dose. Each challenge point denotes 3 vaccinated and 3 control (unvaccinated) camels. The red arrow shows the number of days post-vaccination of challenge study.

**Figure 2 animals-13-01513-f002:**
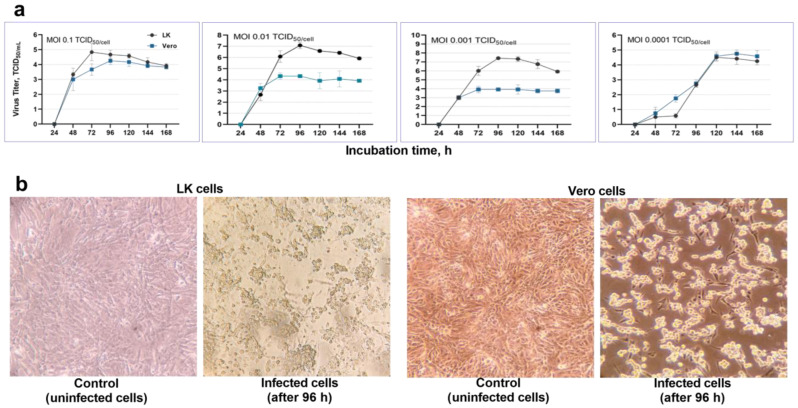
Comparative cultivation of the camelpox virus (CMLV) in cell cultures depending on the multiplicity of infection (MOI). (**a**) Infectious titer of the virus in LK and Vero cell cultures at different MOI. (**b**) Cytopathological effects of the virus on cell cultures monolayers.

**Figure 3 animals-13-01513-f003:**
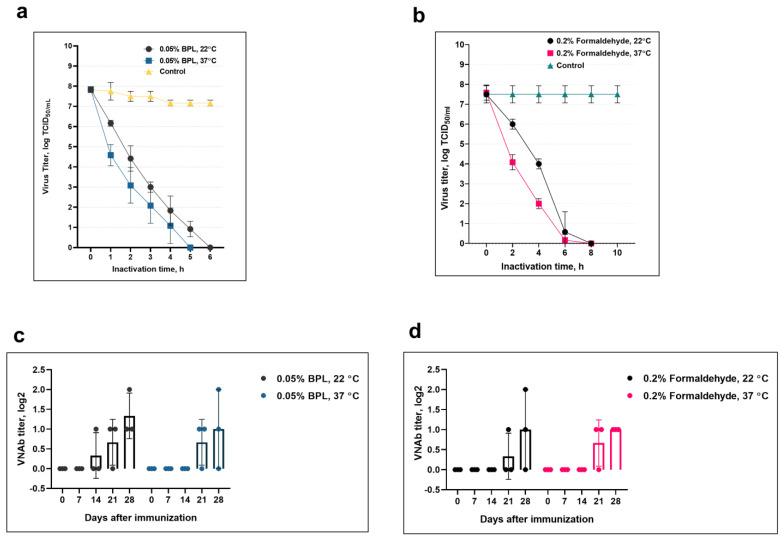
Inactivation of the virus by a chemical method and the persistence of its antigenic activity after inactivation. The top two images show the results of virus inactivation with beta-propiolactone (BPL) (**a**) and formaldehyde (**b**). The bottom two images below show capability to form an immune response in goats inoculated with inactivated material treated with BPL (**c**) and formaldehyde (**d**).

**Figure 4 animals-13-01513-f004:**
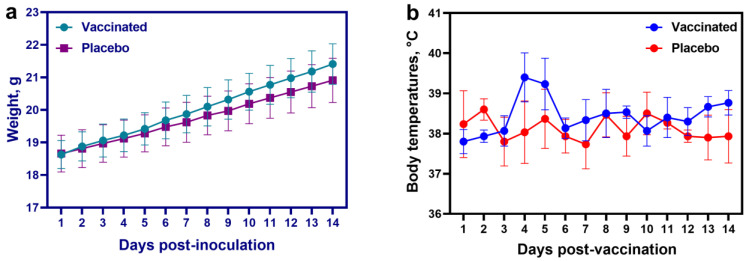
Results of the safety assessment of the experimental inactivated camelpox vaccine in mice and camels. (**a**) Change in body weight of mice after intraperitoneal inoculation of the inactivated vaccine. (**b**) Temperature reaction of camels to the introduction of an experimentally inactivated vaccine.

**Figure 5 animals-13-01513-f005:**
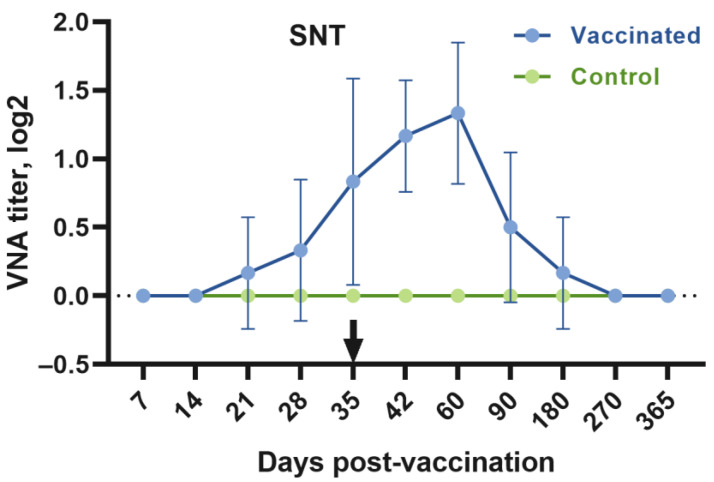
Dynamics of virus-neutralizing antibodies (VNAs) formation to CMLV in six animals immunized with the inactivated adjuvant camelpox vaccine after the first and second vaccine doses. The arrow indicates the number of days post-vaccination of the second dose of vaccine. Data are shown as mean ± standard error of the mean (SEM) of six animals in the vaccinated group.

**Figure 6 animals-13-01513-f006:**
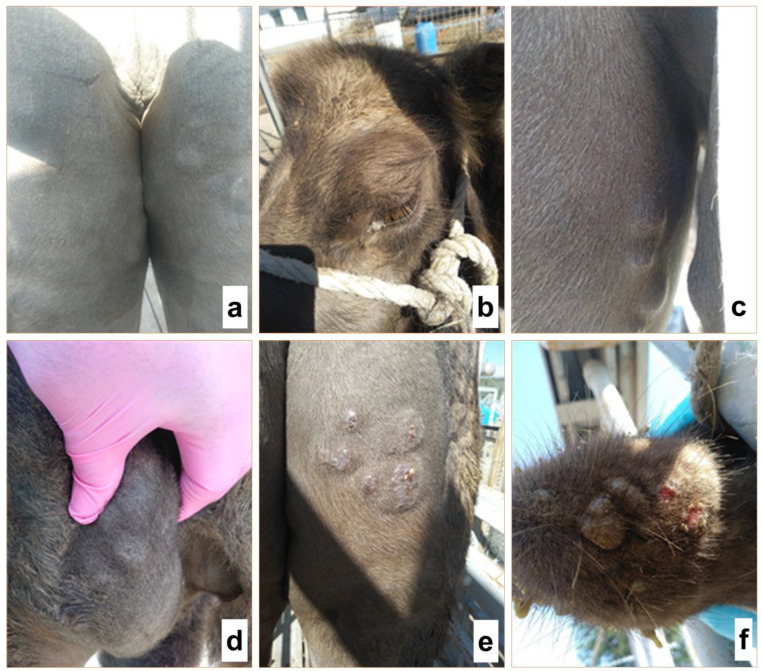
Clinical signs of camelpox in unvaccinated camels after challenge with the virulent M-96 strain. (**a**) Appearance of camelpox papules at the site of virus injection, 4 days; (**b**) purulent discharge from the eyes, 5 days; (**c**) camelpox papules, 7 days; (**d**) prescapular lymph node increased, 9 days; (**e**) formation of crusts at the injection site and generalization of the camelpox process, 12 days; and (**f**) appearance of camelpox on the lower lip, 14 days.

**Table 1 animals-13-01513-t001:** Delayed type hypersensitivity test (DTH) in camels with inactivated vaccine after on the 60th, 180th, and 365th DPV.

Days Post-Vaccination	Camel Identification Number	Skin Thickness (mm), Days after Inoculation					
		0	1	2	3	4	5
60 DPV	009 (vaccinated)	1.5	3.5	4.5	5.2	5.0	5.0
	010 (vaccinated)	1.2	1.5	2.1	3.4	5.1	5.4
	005 (unvaccinated)	1.5	1.5	1.5	1.5	1.5	1.5
180 DPV	011 (vaccinated)	1.0	2.0	2.2	2.9	2.5	2.4
	012 (vaccinated)	1.5	2.7	2.9	2.8	2.5	2.5
	007 (unvaccinated)	1.3	1.3	1.3	1.3	1.3	1.3
365 DPV	013 (vaccinated)	1.1	2.5	2.9	2.7	2.4	2.4
	015 (vaccinated)	1.5	2.9	2.7	2.5	2.1	2.3
	006 (unvaccinated)	1.3	1.3	1.3	1.3	1.3	1.3

## Data Availability

The data obtained during this study are openly available in the KNB Data Repository at doi:10.5063/F1GM85RZ.

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
