# Peer review of "Development of an Inactivated Camelpox Vaccine from Attenuated Camelpox Virus Strain: Safety and Protection in Camels"

_animals, 2023, doi:10.3390/ani13091513_

Round 1
Reviewer 1 Report (Previous Reviewer 1)
The manuscript has been improved by taking into account comments in my previous review.
Author Response
We are glad that we could address the reviewers’ concerns raised during the revision of this manuscript.
Reviewer 2 Report (Previous Reviewer 2)
This study describes the safety and protection efficacy of an attenuated camelpox vaccine.
Authors have done extensive experiments to prove their hypothesis; however, I have a few concerns that should be addressed, particularly they need to work on the written presentation of the results.
Line 59-61: Authors mention that four vaccines are available, and two have been commercialized. Have these vaccines been used in this region or other parts of Kazakhstan? Why exactly do we need to have another vaccine? Provide the drawbacks of already available vaccines here. The authors need to describe this paragraph by mentioning the disadvantages of the vaccine first and then coming towards the objectives of this study.
Lines 115-118: Authors have used two different reagents to inactivate the virus. Is there a particular reason for using these chemicals? Why did they not just use high temperatures to inactivate the virus? Why did they not include this as an additional method in their study? It would have been interesting to see if there is any difference, at least in their in vitro studies.
Lines 85-87: Titers of the virulent strain of CMLV have yet to be mentioned here.
Lines 113-115: Autors mentioned that they used 1000 rpm; I suggest they use high-speed centrifugation for their virus stocks in future. I understand that cells settle down at speed used here, but high-speed centrifugation ensures all the cells in the supernatant settle down before the stocks are prepared.
Figure 1 is confusing and does not correctly represent the number of experimental animals. The authors must point out both times that they used three control and three vaccinated animals.
Section 2.8: It is confusing here; the titers mentioned above represent EID50, and they say it in TCID50. They need to clarify how the virus was titrated. Previously, some studies have shown that there could be a difference ins sensitivity of the virus based on the titration methods used. Was the same virus stock used, and was some formula used to convert EID50 to TCID50?
Line 180: Why do they mention 3-5 x 104 cells/well here? There is a big range; how do they count the number of cells in the lab? They may make a slight difference here, but ensuring that a uniform number of cells are used to perform such sensitive assays is crucial.
Line 180: Replace moist atmosphere with humidified and correct throughout.
Lines 224-227: Rewrite. You mentioned the acronym for CPE multiple times here, and check this mistake throughout the manuscript.
Line 227: At the highest dose (multiplicity of 227 infections [MOI] 0.1 TCID50), this is not the correct way to mention MOI, do not put TCID50 values here. Just an MOI of 0.1 is sufficient to understand.
Results section 3.1 needs to be clarified; it is hard to understand the authors' voice here. Authors must rewrite the whole paragraph. Some suggestions are again to start with the strain of the virus used and keep in mind to mention the cell line, and there is no need to present complex statistical data here. Please feel free to use this as a supplementary table if you need to. Could you provide the p values in the text only? Also, section 2.4 of the materials and methods details how these cells were infected. How long were they left in the incubator before infection etc.?
Figure 4: Results of the safety assessment of the experimental inactivated camelpox vaccine in 294 mice and camels. This should be presented differently. Rather than showing data for the individual animals, authors may show a mean with whiskers/bars to clarify it. Additionally, they may provide the detailed results as a supplementary table if they want. Same as in figure 5.
Section 3.4 of results: Replaced antibodies were formed with detected.
Discussion: authors may discuss more the previous studies on live vaccines here to justify their claims.
Round 2
Reviewer 2 Report (Previous Reviewer 2)
Line 125-131: The revised text contains poor technical language that requires improvement. To address this issue, authors may refer to other papers and rewrite the entire paragraph. For example, when the cell monolayer was damaged is not technically correct.
Line 248-257: Authors have simply deleted unnecessary text and have not made any improvements to the results section. It is important for them to present the results shown in Figure 2 clearly.
The authors should provide a clear description of how they calculated the antibody titers in the Methods section, and in the Results section, they should discuss whether they believe that such low neutralizing antibody titers may provide protection against the virus.
Throughout the paper, the authors have used inconsistent capitalization for 'Camelpox' (sometimes capitalized, sometimes not). They need to ensure a uniform pattern is used.
The technical language used throughout the manuscript requires improvement, as previously recommended during the revision process.
Round 3
Reviewer 2 Report (Previous Reviewer 2)
The authors have made significant improvements to the article.
This manuscript is a resubmission of an earlier submission. The following is a list of the peer review reports and author responses from that submission.
Round 1
Reviewer 1 Report
Zhugunissov et al have carried out experiments to investigate the possibility of vaccinating camels against camelpox virus infection with the homologous virus that has been inactivated with chemical agents. The authors are well aware that poxvirus vaccines are usually attenuated strains because these strains induce the broadest kind of immune response, both humoral and cellular. Nevertheless, there are concerns among farmers as well as presumably vaccine manufacturers that an attenuated strain may revert or already contain virulent material that could be the cause of a camelpox virus outbreak. Thus, the authors have attempted once again, as has often been attempted in the past, to use an inactivated virus for poxvirus vaccination. Overall, their procedure has been successful and the methods, that are standard in virology, appear promising for the development of a killed camelpox virus vaccine. Nevertheless, the manuscript and the description of the procedures used have a number of shortcomings.
Major points
In particular, the authors do not provide a detailed description of the way inactivation was carried out. Presumably the virus preparation before inactivation is a crude infected cell lysate but this is not mentioned. How was beta propiolactone or formadehyde removed from the inactivated virus before use or if not removed this should be stated. It’s mentioned on lines 102-103 that the virus was clarified by centrifugation. The sentence implies that virus was recovered in the supernatant so at what speed in “g” and for how long was centrifugation carried out? What is in the pellet?
Another essential item that needs clarification is how the level of serum neutralization was determined. The authors mention a neutralization index but fail to provide the precise way this index was calculated. The formula should be provided and they must indicate if they assayed for 50% or 90% (or some other level) of reduction in titre? In any case the level of neutralizing antibodies appears to be extremely weak as the authors recognize in the discussion section but fail to mention in the results section.
Concerning the ELISA test the authors have not calculated the level of antibodies according to the way this is calculated by Abbexa in their “instructions for use” which mentions that serum samples are considered positive if the values are greater than the value of the negative control added to 0.15. This implies that their ELISA results are slightly underestimated.
The statement lines 175-176 that CPE of the virus was also different in both cells… does not appear accurate from the figure provided. Both cells types appear to undergo similar cell aggregation. There may be an appearance of a difference due to the use of phase contrast microscopy for one cell line and not the other.
A key finding of the manuscript is the protection of the vaccinated camels however the authors fail to mention how many vaccinated animals and control animals were challenged. More details on these experiments than those in figure 5 are necessary for readers to be convinced that the inactivated vaccine actually protects camels from a challenge infection.
Minor points
The authors use either CMLP or CMLPV as an abbreviation for camelpox virus. They should use the same abbreviation throughout the manuscript either CMLP or preferably CMLV, the latter being the abbreviation recommended by the International Committee on the Taxonomy of Viruses.
Line 18 : instead of “sorbed” the authors probably mean “adsorbed” but is the aluminium hydroxide actually adsorbed or simply mixed with the inactivated virus?
Lines 34-35 “which includes humans, other mammals, fish, reptiles and invertebrates” instead of “which includes human, animals, birds and insects”.
Lines 37-38 “weakness” instead of “loss of condition”
Line 71 “in chicken embryos” instead of “in a chicken embryo”
Line 98 “stationary method”. I don’t think this is the adequate term. What do the authors mean by “stationary method”?
Figure 1b does not provide any useful information and can be removed
Figure 2 needs to be mentioned in paragraph 3.2.
In paragraph lines 207-213 the authors mention that the BPL inactivated virus appears to induce an earlier immune response that the formaldehyde inactivated virus but that this difference is not statistically significant. They then state that based on their results the BPL inactivated virus was chosen for further studies. This latter statement is not warranted since the difference was not significant.
Line 285 do the authors mean “scab’ when they use the word “seal”?
Lines 337-338 What does the sentence “In this regards, we tried to determine the “walking” mode of inactivation of the CMLPV…”
Line 359 what is the “presence of tension”?
Author Response
Zhugunissov et al have carried out experiments to investigate the possibility of vaccinating camels against camelpox virus infection with the homologous virus that has been inactivated with chemical agents. The authors are well aware that poxvirus vaccines are usually attenuated strains because these strains induce the broadest kind of immune response, both humoral and cellular. Nevertheless, there are concerns among farmers as well as presumably vaccine manufacturers that an attenuated strain may revert or already contain virulent material that could be the cause of a camelpox virus outbreak. Thus, the authors have attempted once again, as has often been attempted in the past, to use an inactivated virus for poxvirus vaccination. Overall, their procedure has been successful and the methods, that are standard in virology, appear promising for the development of a killed camelpox virus vaccine. Nevertheless, the manuscript and the description of the procedures used have a number of shortcomings.
Major points
- In particular, the authors do not provide a detailed description of the way inactivation was carried out. Presumably the virus preparation before inactivation is a crude infected cell lysate but this is not mentioned. How beta propiolactone or formaldehyde was removed from the inactivated virus before use or if not removed this should be stated. It’s mentioned on lines 102-103 that the virus was clarified by centrifugation. The sentence implies that virus was recovered in the supernatant so at what speed in “g” and for how long was centrifugation carried out? What is in the pellet?
AU response: It is now revised as requested in lines 112-117
- Another essential item that needs clarification is how the level of serum neutralization was determined. The authors mention a neutralization index but fail to provide the precise way this index was calculated. The formula should be provided and they must indicate if they assayed for 50% or 90% (or some other level) of reduction in titre? In any case the level of neutralizing antibodies appears to be extremely weak as the authors recognize in the discussion section but fail to mention in the results section.
AU response: Authors would like to declare that neutralization index was not calculated in this study. It must be a technical error occurred while writing up this manuscript. We did calculate neutralizing titer instead. It is now indicated in lines 169-172.
- Concerning the ELISA test the authors have not calculated the level of antibodies according to the way this is calculated by Abbexa in their “instructions for use” which mentions that serum samples are considered positive if the values are greater than the value of the negative control added to 0.15. This implies that their ELISA results are slightly underestimated.
AU response: Regarding the ELISA test, we included the ELISA results as an additional study. In fact, the ELISA results do not show any data and are not informative. In addition, we recently discovered that this test was not validated. In this regard, we unanimously decided to remove the results of the ELISA in order not to mislead readers..
- The statement lines 175-176 that CPE of the virus was also different in both cells… does not appear accurate from the figure provided. Both cells types appear to undergo similar cell aggregation. There may be an appearance of a difference due to the use of phase contrast microscopy for one cell line and not the other.
AU response: We agree with reviewer’s comments. We observed that at final point, both cell lines gets aggregated in similar pattern. However what we’ve noticed is that Vero cells get aggregated faster than that LK cell line.
- A key finding of the manuscript is the protection of the vaccinated camels however the authors fail to mention how many vaccinated animals and control animals were challenged. More details on these experiments than those in figure 5 are necessary for readers to be convinced that the inactivated vaccine actually protects camels from a challenge infection.
AU response: It is now revised as requested in lines 142-145.
Minor points
- The authors use either CMLP or CMLPV as an abbreviation for camelpox virus. They should use the same abbreviation throughout the manuscript either CMLP or preferably CMLV, the latter being the abbreviation recommended by the International Committee on the Taxonomy of Viruses.
AU response: Revised as requested.
- Line 18 : instead of “sorbed” the authors probably mean “adsorbed” but is the aluminium hydroxide actually adsorbed or simply mixed with the inactivated virus?
AU response: It is now revised as requested in line 26.
- Lines 34-35 “which includes humans, other mammals, fish, reptiles and invertebrates” instead of “which includes human, animals, birds and insects”.
AU response: It is now revised as requested in line 43.
- Lines 37-38 “weakness” instead of “loss of condition”
AU response: It is now revised as requested in line 46.
- Line 71 “in chicken embryos” instead of “in a chicken embryo”
AU response: It is now revised as requested in line 80.
- Line 98 “stationary method”. I don’t think this is the adequate term. What do the authors mean by
“stationary method”?
- AU response: It is now revised as indicated in line 109.
- Figure 1b does not provide any useful information and can be removed
AU response: Revised as requested
- Figure 2 needs to be mentioned in paragraph 3.2.
AU response: Revised as requested
- In paragraph lines 207-213 the authors mention that the BPL inactivated virus appears to induce an earlier immune response that the formaldehyde inactivated virus but that this difference is not statistically significant. They then state that based on their results the BPL inactivated virus was chosen for further studies. This latter statement is not warranted since the difference was not significant.
AU response: Yes, I agree with your comment, and we have covered not only this feature, but also other benefits of beta propiolactone. For example, Beta-propiolactone inactivated the virus 2 hours earlier than formaldehyde. In addition, beta-propiolactone easily and quickly decomposes after 24 hours with the formation of harmless substances: hydroacrylic and beta-hydroxypropionic acids.
- Line 285 do the authors mean “scab’ when they use the word “seal”?
AU response: It is now revised as indicated in line 309.
- Lines 337-338 What does the sentence “In this regards, we tried to determine the “walking” mode of inactivation of the CMLPV…”
AU response: It is now revised as indicated in line 363.
- Line 359 what is the “presence of tension”?
- AU response: It is now revised as indicated in line 385.

Reviewer 2 Report
This paper describes the development of an inactivated vaccine for attenuated Camel pox virus. Currently, live attenuated vaccine is used in some parts of Kazakhstan to contain the virus, but there have been complaints from the farmers about the safety concerns of the vaccine. Therefore, the authors worked on the development of inactivated vaccines. They titrated the virus in LK and Vero cells and showed that LK cells are highly susceptible and permissive to the CMLP virus. Then, they showed that beta propiolactone is an effective virus inactivation agent. The vaccine proved to be safe in mice and camels. Serum-neutralization antibodies were detected 42 days after the second dose of vaccination.
Similarly, antibodies were also detected in ELISA at 42 days after vaccination. In response to the challenge with the M-96 strain of the CMLP virus, vaccinated animals did not present any clinical signs, while unvaccinated animals got seriously sick. Furthermore, they showed that even in the absence of antibodies, vaccinated animals did not develop any clinical symptoms indicating that cellular immune response may be protective in case of infection.
Overall, the study has been well designed, and all the experiments have been conducted to show the protection and safety of an inactivated vaccine.
Authors should work to improve the technical language of the paper and use appropriate technical terms to describe the results.
- Include statistical analysis in the methods section for each part of the study. Also, put details of the tools used to generate figures etc.
- Figure 1A: Use the same scale (Y-axis) to elaborate on the difference at different MOIs.
- Line 197: CPE cell culture monolayer? Authors may use better terminology to present their results. Do you mean to say no CPE was detected?
- Fig 2A, B: Use a different acronym for formaldehyde (FD); the form is confusing and not standard.
- VNA formation is not the correct term.
- Figure 4: Make the figure uniformly. See the legends SNT and ELISA; use the same fonts.
- Do the authors know the reason behind the high background in their ELISA? High ODs were detected in their control samples. How have they established a cut-off for positivity? Provide all the details in the methods section.
- The results provide evidence about the importance of cell-mediated immunity but why they have not included any assay to assess cellular immune response. PBMC stimulation would be the best one in this case.
- Authors may also present their experimental timeline alongside the figures.
Author Response
This paper describes the development of an inactivated vaccine for attenuated Camel pox virus. Currently, live attenuated vaccine is used in some parts of Kazakhstan to contain the virus, but there have been complaints from the farmers about the safety concerns of the vaccine. Therefore, the authors worked on the development of inactivated vaccines. They titrated the virus in LK and Vero cells and showed that LK cells are highly susceptible and permissive to the CMLP virus. Then, they showed that beta propiolactone is an effective virus inactivation agent. The vaccine proved to be safe in mice and camels. Serum-neutralization antibodies were detected 42 days after the second dose of vaccination.
Similarly, antibodies were also detected in ELISA at 42 days after vaccination. In response to the challenge with the M-96 strain of the CMLP virus, vaccinated animals did not present any clinical signs, while unvaccinated animals got seriously sick. Furthermore, they showed that even in the absence of antibodies, vaccinated animals did not develop any clinical symptoms indicating that cellular immune response may be protective in case of infection.
Overall, the study has been well designed, and all the experiments have been conducted to show the protection and safety of an inactivated vaccine.
Authors should work to improve the technical language of the paper and use appropriate technical terms to describe the results.
- Include statistical analysis in the methods section for each part of the study. Also, put details of the tools used to generate figures etc.
AU response: Statistical analysis has been added to the M&M section, see lines 191-197.
- Figure 1A: Use the same scale (Y-axis) to elaborate on the difference at different MOIs.
AU response: Revised as requested.
- Line 197: CPE cell culture monolayer? Authors may use better terminology to present their results. Do you mean to say no CPE was detected?
AU response: It is now revised as indicated in line 231.
- Fig 2A, B: Use a different acronym for formaldehyde (FD); the form is confusing and not standard.
AU response: It is now revised as indicated in line 253.
- VNA formation is not the correct term.
AU response: We understand the concern of the respected reviewer, however the term “formation of VNA” is widely used in other works published in decent journals. We will, however, yield to the decision of the editor in this regard.
- Figure 4: Make the figure uniformly. See the legends SNT and ELISA; use the same fonts.
AU response: Revised as requested. Regarding the ELISA test, we included the ELISA results as an additional study. In fact, the ELISA results did not show any data and are not informative. In addition, we recently discovered that this test was not validated. In this regard, we unanimously decided to remove the results of the ELISA in order not to mislead readers.
- Do the authors know the reason behind the high background in their ELISA? High ODs were detected in their control samples. How have they established a cut-off for positivity? Provide all the details in the methods section.
AU response: Again, we unanimously decided to remove the results of the ELISA because it turned out that this particular ELISA test hasn’t been validated.
- The results provide evidence about the importance of cell-mediated immunity but why they have not included any assay to assess cellular immune response. PBMC stimulation would be the best one in this case.
AU response: Unfortunately, we did not study cellular immunity due to the lack of reagents, materials, and equipment needed to study cellular immunity, including PBMC stimulation.
- Authors may also present their experimental timeline alongside the figures.
AU response: We described the schedule of experiments in Section 2.7. Challenge study. To avoid repetitions, we think that it is not necessary to indicate the schedule of experiments along with the figures.

Round 2
Reviewer 1 Report
The authors have made some changes to their manuscript that satisfy most of the minor questions raised previously in my review however some major issues still remain. For instance, the manner in which neutralizing antibodies was determined is not presented without enough detail and in imprecise repetitive language as follows on lines 168-174: “The neutralizing antibody titer was represented as damage to the cell monolayer in the wells when viewed under a microscope,. The neutralizing titer of the studied sera was considered if characteristic CPE of the cell monolayer was observed in the well when compared with control sera. The neutralizing antibody titer was represented as damage to the cell monolayer in the wells when viewed under a microscope,. The neutralizing titer of the studied sera was considered if no CPE of the cell monolayer was observed in the well when compared with control sera.”
The challenge studies are still not presented in detail as requested in my previous review. Apparently 9 animals were vaccinated once for 2 of them and twice for the others. Then the animals were challenged at different times points with two animals for each of the four time points. So apparently there’s an extra animal that was included in one of the time points but is not mentioned. The authors limit their discussion to mentioning that all animals were protected from the challenge whether with or without a booster vaccination. Their discussion of the challenge study is focussed on the immune response with simply a general statement on the protection of all vaccinated animals whatever the protocol used. (“Thus, after analyzing the studies, it was found that the early period of a partial immune response in camels immunized with an inactivated vaccine began on the 30th day after the first immunization, but a 100% immune response occurred on the 21st day after the second vaccination. When studying the duration of immunity created by the inactivated vaccine, specific antibodies to the CMLV were detected in the blood of vaccinated animals on the 60th days after the second immunization, a finding that confirms the presence and duration of humoral immunity in the body of immunized animals.”). A nearly identical statement is repeated in the discussion.
Further the results presented in figure 4 which indicate that neutralizing antibodies were determined on 6 animals contradict the experimental setup which uses only 2 animals per group.